# Global antimicrobial resistance and use surveillance system (GLASS 2022): Investigating the relationship between antimicrobial resistance and antimicrobial consumption data across the participating countries

**Samuel Ajulo**[ID]**, Babafela Awosile**[ID]*

School of Veterinary Medicine, Texas Tech University, Amarillo, Texas, United States of America

* babafela.awosile@ttu.edu

## Abstract

For the first time since 2015, the World Health Organization's (WHO) global Antimicrobial Resistance and Use Surveillance (GLASS) featured both global reports for antimicrobial resistance (AMR) and antimicrobial consumption (AMC) data in its annual reports. In this study we investigated the relationship of AMR with AMC within participating countries reported in the GLASS 2022 report. Our analysis found a statistically significant correlation between beta-lactam/cephalosporin and fluoroquinolones consumption and AMR to these antimicrobials associated with bloodstream *E. coli* and *Klebsiella pneumoniae* among the participating countries (P<0.05). We observed that for every 1 unit increase in defined daily dose DDD of beta-lactam/cephalosporins and quinolone consumptions among the countries, increased the recoveries of bloodstream-associated beta-lactam/cephalosporins-resistant *E. coli*/*Klebsiella* spp. by 11–22% and quinolone-resistant *E. coli*/*Klebsiella* spp. by 31–40%. When we compared the antimicrobial consumptions between the antimicrobial ATC (Alphanumeric codes developed by WHO) groups and countries, we observed a statistically significant higher daily consumption of beta-lactam-penicillins (J01C, DDD difference range: 5.23–8.13) and cephalosporins (J01D, DDD difference range: 2.57–5.13) compared to other antimicrobial groups among the countries (adjusted for multiple comparisons using Tukey's method). Between the participating countries, we observed a statistically significant higher daily consumption of antimicrobial groups in Iran (DDD difference range: 3.63–4.84) and Uganda (DDD difference range: 3.79–5.01) compared to other participating countries (adjusted for multiple comparisons using Tukey's method). Understanding AMC and how it relates to AMR at the global scale is critical in the global AMR policy development and implementation of global antimicrobial stewardship.

**Data Availability Statement:** Data for this project can be access on the World Health Organization's

Global Antimicrobial Resistance and Use Surveillance System (GLASS) report published for 2022[20].

**Funding:** The authors did not receive any grant funding for this project.

**Competing interests:** The authors do not have any conflict of interest.

## Introduction

Antimicrobial resistance (AMR) is a complex and rapidly evolving challenge that threatens the very foundation of modern medicine and healthcare [1, 2]. The consequences of AMR are staggering, in 2019 alone, AMR was estimated to have been responsible for about 1.3 million deaths globally and it was projected to rise to 10 million yearly by 2050 if nothing was done to address it [1, 3]. Aside from the deaths due to AMR, it has significantly resulted in increased costs of health care and services, for example, in the United States alone, more than $4.6 billion in health care costs annually was attributed to treating six alarming AMR threats [4, 5]. Also, the economic implication of AMR is on the high side, the World Bank Group simulated about $1.1 trillion annual global economic loss by 2030 projected to reach $2 trillion annually by 2050 in an optimistic low-AMR scenario [6].

The development of antimicrobial resistance is accelerated by selective pressure exerted by using and misusing antimicrobials [7]. From an evolutionary perspective, the presence of an antimicrobial agent provides a selective pressure for AMR development, however, injudicious use which includes misuse and over-use of antimicrobials rapidly drives the selective pressure for AMR development [3, 7–10]. For human health, several factors such as consumer or patient demand for antimicrobials, availability and access to antimicrobials, knowledge about judicious use of antimicrobials, poor disease diagnosis, and inferior quality of antimicrobials amongst several others have been reported as key drivers of antimicrobial misuse and over-use [11–14]. Over the years with the introduction of new antimicrobials and increased use, there has been increasing development and dissemination of multidrug-resistant (MDR) bacterial pathogens, such as MDR *Mycobacterium tuberculosis*, and several MDR hospitals-acquired infections like *E. coli*, *Acinetobacter baumannii*, *Klebsiella pneumoniae*, *Staphylococcus aureus* and *Streptococcus pneumoniae* [15, 16]. Because of the persistent challenge of AMR, some anti-microbials have been classified by WHO as critically important to human medicine to tackle AMR, and within this group, certain antimicrobials considered as the highest priority to treat MDR bacterial infection are quinolones, third and fourth-generation cephalosporins, macro-lides and ketolides, and glycopeptides [17, 18]. Also, under the WHO's AWaRe (Access, Watch, and Reserve) classification which was developed for evaluation and monitoring the use of antimicrobials, these critically important antimicrobials mostly belong to the Watch and Reserve list which are antimicrobial with an elevated risk of bacteria resistance selection and the last options for human treatment, respectively [17, 19]. However, in recent times there have been increasing reports of resistance against these highest-priority antimicrobials [20–22].

In the face of the rapid development of antimicrobial resistance, the surveillance of both antimicrobial consumption (AMC) and antimicrobial resistance (AMR) are critical tools for understanding the dynamics of antimicrobial resistance and play a key role in the development of AMR stewardship programs and policy [23–25]. There have been huge efforts at the national, regional, and global levels toward AMR/AMC surveillance. For example, the European Antimicrobial Resistance Surveillance Network (EARS-Net) is the largest regional sur-veillance network in the world consisting of collaborative efforts of 29 countries [26]. Although there has been a constant effort targeted towards the improvement of the various regional and national surveillance systems, however, several surveillance systems are still faced with challenges of scarcity of accurate and reliable information, lack of formal framework for data collection and sharing between laboratory networks, and a limited number of enrolment by individual countries in the regional surveillance, amongst several others, especially in the low-income countries (LICs) low and middle-income countries (LMICs) [21, 27, 28]. Because AMR is a global threat and can easily spread across the world, surveillance on a global scale

was therefore pertinent to tackle AMR and to strengthen several national and regional surveillance systems [27]. In 2015, the WHO announced its first global collaborative effort to improve and regulate AMR surveillance as one of the pillars of its global action plan (GAP) for AMR evidence base which is the Global Antimicrobial Resistance and Use Surveillance System (GLASS) [28]. In its early implementation between 2015 and 2019, it started with only monitoring and reporting AMR rates and trends in common bacteria with 42 countries enrolled by the end of the first data call in 2017, which later expanded to 126 in the first data call in 2020 [20, 28]. However, global AMC data was reported independently, and the first global AMC data was published in 2018 with 65 participating countries [29]. AMR and AMC data were reported together for the first time in the 2022 GLASS report [20].

In the 2022 GLASS report, WHO highlighted elevated levels of AMR, particularly in the bloodstream causing infection with higher levels of resistance to third-generation cephalosporins *in Klebsiella pneumoniae*, the third most prevalent pathogen that causes bloodstream infections which in turn has increased the use of last resort carbapenems and a world-wide spread of carbapenems-resistance *Enterobacterales* and high rates of carbapenem and aminoglycoside resistance in Acinetobacter spp which are of grave concern because carbapenem-resistant strains are often MDR and have been associated with treatment failures [20, 22, 30]. These resistance levels were strongly associated with the use of antimicrobials driven by several factors in different countries and regions [11, 12, 31–33]. Being the first time the GLASS report will be integrating the reports of AMR data with AMC data, the objective of this study was to associate the reported AMR data with the AMC data in different countries and regions based on the reported GLASS data, focusing on beta-lactam/cephalosporin and quinolones AMC data and beta-lactam/cephalosporin and quinolones resistant *E. coli* and *Klebsiella pneumoniae* and also explore the relationship and differences between the AMC data reported among the participating countries. We have particularly focused this study on *E. coli* and *Klebsiella pneumoniae* because of the WHO GLASS 2022 highlight of elevated resistance levels in bloodstream infection which is highly fatal and *K. pneumoniae* is mostly implicated [20]. Also, *E. coli* generally serves as one of the most important indicator organisms for AMR surveillance and monitoring. With these, we are more likely to see a more distinct relationship between AMU and AMR [34, 35]. This analysis would help gain better insight into GLASS data and provide a future reference for integrated analysis of AMR and AMC data at a global scale.

## Materials and methods

### Data source and analysis

The data used in this study were extracted from the GLASS report published for 2022. GLASS is an international collaboration under the World Health Organization devoted to the collection, integration, and presentation of data on antimicrobial resistance (AMR) and antimicrobial consumption (AMC) from the national surveillance systems of the participating countries. This framework aimed to harmonize the reporting of quality and representative data of AMR and AMC on a global scale. While 216 countries, territories, and areas (CTAs) are enrolled in the GLASS program, data availability varies from country to country. However, data on AMC were available for 26 CTAs for the year 2020. Data on AMC were provided by each country for different classes of antimicrobials for systemic use using the alphanumeric codes (ATC codes) developed by the World Health Organization (WHO) for the classification of drugs and other medical products. The Antimicrobial consumption data were provided by GLASS both in adjusted and unadjusted measurements, however, for this study, we extracted the AMC data expressed as a defined daily dose (DDD) adjusted by the population size of each participating country for Beta-lactam antibacterials, penicillins group (J01C), Other beta-lactam

antibacterials group (J01D), and Quinolone antibacterials group (J01M). The DDD was presented as the number of DDDs per 1000 inhabitants per day, which is interpreted as the average number of individuals per 1000 inhabitants on antimicrobial treatment each day. For the AMR data, we extracted data on bloodstream infection for beta-lactam and fluoroquinolone-resistant *Escherichia coli* and *Klebsiella pneumoniae*. The AMR data were presented as percentage-resistant bacteria isolated from total bacteriologically confirmed infections. For both AMR and AMC data, our analyses focused on the data for the year 2020.

We used Microsoft Excel for data management. Data were subsequently imported into the R software environment for analysis using different R packages. We explored the linear relationship between the AMR and AMC data reported among the countries using Spearman's correlation analysis. We also explored the effect of the antimicrobial consumption reported among the participating countries on the prevalence of beta-lactam and fluoroquinolone-resistant *E. coli* and *Klebsiella pneumoniae* using a mixed-effect regression model with the beta distribution. We further used a multivariable linear regression model to explore the relationship and differences in the antimicrobial consumption data (DDD) reported among the participating countries. In this linear model, we used the DDD per 1000 inhabitants per day as the outcome variable while the country and antimicrobial drug ATC group were predicting variables. We compared the DDD between the countries and the DDD between the antimicrobial drug ATC groups. We adjusted for multiple comparisons using Tukey's method in both cases. Standard regression model diagnostics were explored for all statistical analyses, statistical significance was set at $P < 0.05$, and all analyses were performed using R software (R version 4.2.2).

## Results

From the results of the correlation analysis for beta-lactam resistance and beta-lactam consumption (Fig 1), we observed a statistically significant positive linear association between bloodstream-associated ceftriaxone-resistant *E. coli* (rho = 0.56, P = 0.05), bloodstream-associated ceftazidime-resistant *E. coli* (rho = 0.64, P = 0.017) and beta-lactam drug consumption among the countries. Similarly, we observed a statistically significant positive linear association between bloodstream-associated ceftriaxone-resistant *Klebsiella pneumoniae* (rho = 0.62, P = 0.037), bloodstream-associated ceftazidime-resistant *Klebsiella pneumoniae* (rho = 0.64, P = 0.017) and beta-lactam drug consumption among the countries. For fluoroquinolone resistance and quinolone consumption correlation analysis (Fig 2), we observed a statistically significant positive linear association between bloodstream-associated ciprofloxacin-resistant *E. coli* (rho = 0.85, P = 0.000034), bloodstream-associated levofloxacin-resistant *E. coli* (rho = 0.83, P = 0.0083) and quinolone consumption among the countries. Similarly, we saw a statistically significant positive linear association between bloodstream-associated ciprofloxacin-resistant *Klebsiella pneumoniae* (rho = 0.74, P = 0.0025), bloodstream-associated levofloxacin-resistant *Klebsiella pneumoniae* (rho = 0.72, P = 0.037) and quinolone consumption among the countries. From the correlation plots, there was an obvious grouping of the developing countries and developed countries in terms of the AMC and AMR rates. Developed countries such as European countries (Germany, Denmark, Sweden, United Kingdom, Belgium) are mostly low antimicrobial consumption-low antimicrobial resistance countries compared to other countries that are most high use-high antimicrobial resistance or low use-high antimicrobial resistance groups. These groupings were consistent for beta-lactam and fluoroquinolone antimicrobial classes.

Using the beta-regression model (Table 1), we observed that for every 1 unit increase in DDD of beta-lactam/cephalosporins consumption among the countries, the odds of isolation of bloodstream-associated ceftriaxone-resistant *E. coli* increased by 22% (OR:1.22, 95%CI:

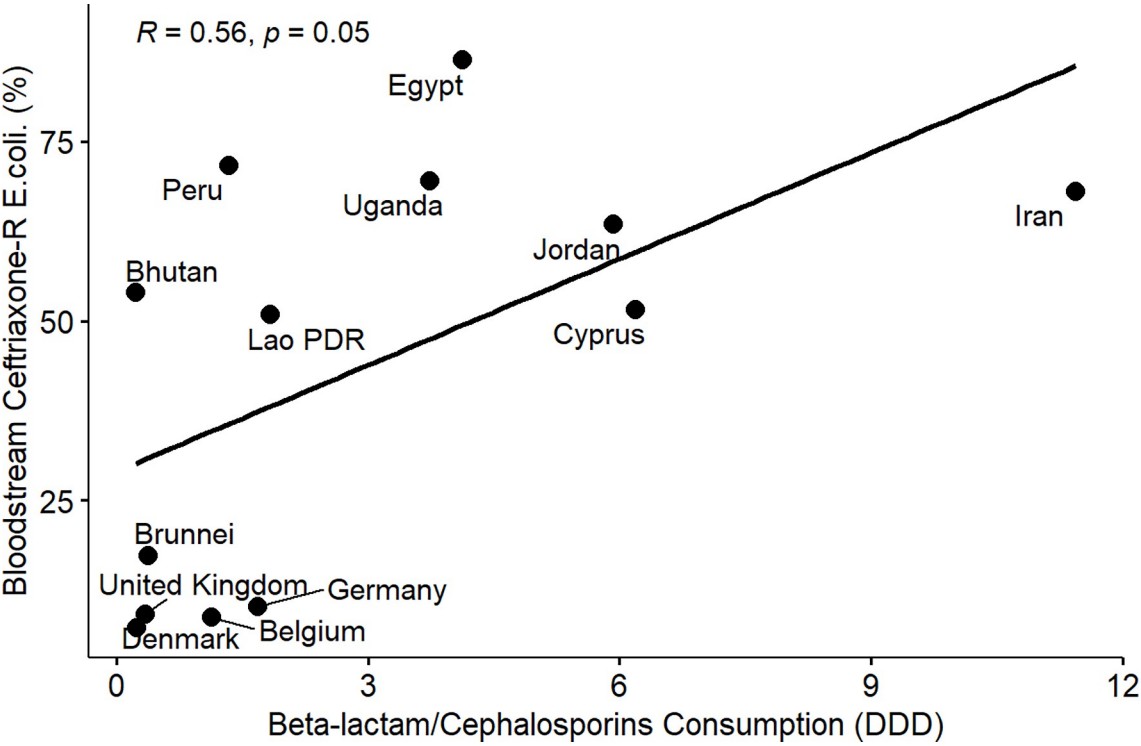

**Fig 1. Linear association and spearman correlation coefficients between bloodstream ceftriaxone-resistant *E. coli* and cephalosporin consumption.**

1.01–1.48), Also, we observed that for every 1 unit increase in DDD of quinolone consumption among the countries, the odds of isolation of bloodstream-associated ciprofloxacin-resistant *E. coli* increased by 40% (OR:1.40, 95%CI: 1.17–1.67), the odds of isolation of bloodstream-associated levofloxacin-resistant *E. coli* increased by 31% (OR:1.31, 95%CI: 1.15–1.50), the odds of isolation of bloodstream-associated ciprofloxacin-resistant *Klebsiella pneumoniae* increased by 37% (OR:1.37, 95%CI: 1.07–1.76), while the odds of isolation of bloodstream-associated levofloxacin-resistant *Klebsiella pneumoniae* increased by 31% (OR:1.31, 95%CI: 1.02–1.68). When we compared the antimicrobial consumptions between the antimicrobial ATC groups (Table 2), and countries (Table 3), we observed a statistically significant higher daily consumption of beta-lactam-penicillins (J01C, DDD difference range: 5.23–8.13) and cephalosporins (J01D, DDD difference range: 2.57–5.13) compared to other antimicrobial groups among the countries (adjusted for multiple comparisons using Tukey's method). Between the participating countries, we observed a statistically significant higher daily consumption of antimicrobial groups in Iran (DDD difference range: 3.63–4.84) and Uganda (DDD difference range: 3.79–5.01) compared to other participating countries (adjusted for multiple comparisons using Tukey's method).

## Discussion

Our study investigated the relationship between antimicrobial resistance (AMR) and antimicrobial consumption (AMC) using data from the GLASS report published for 2022. We found a statistically significant positive linear association between extended-spectrum cephalosporin (ceftriaxone and ceftazidime) and fluoroquinolone (ciprofloxacin and levofloxacin) -resistant *E. coli* and *Klebsiella pneumoniae* isolated from bloodstream infections and the consumption

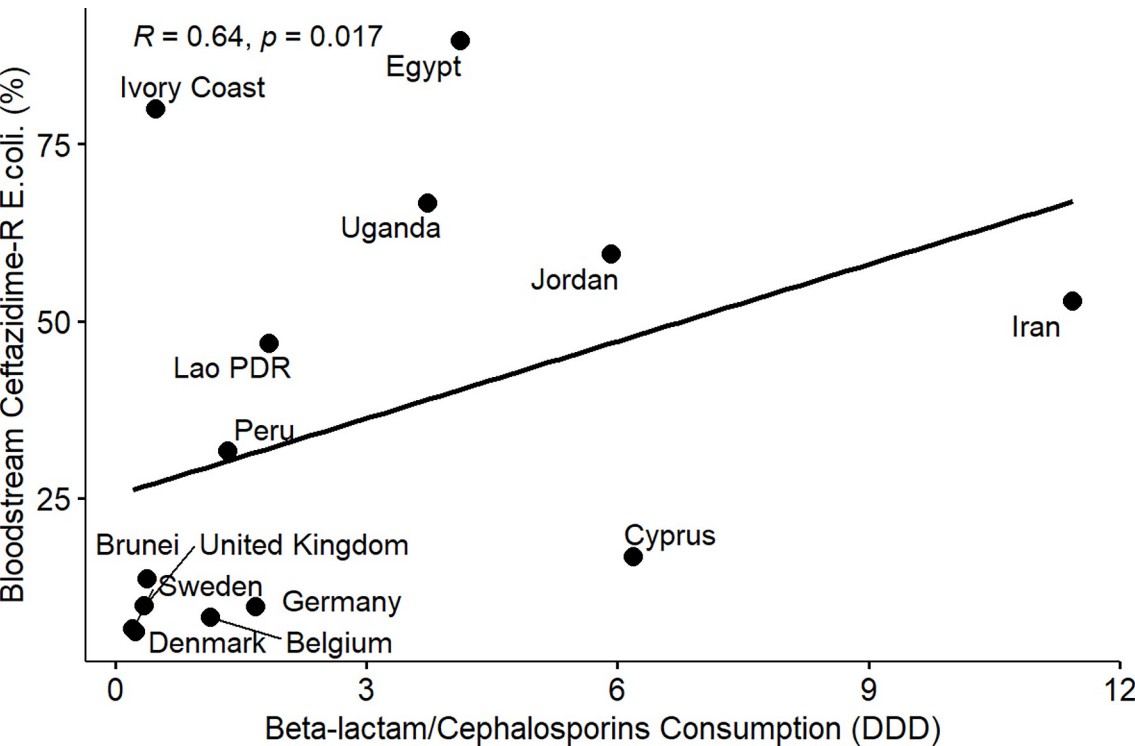

**Fig 2. Linear association and spearman correlation coefficients between bloodstream ceftazidime-resistant *E. coli* and cephalosporin consumption.**

of beta-lactam/cephalosporins and quinolones among the participating countries. These findings are consistent with previous research works that have demonstrated the strong association between antibiotic consumption and the emergence of antibiotic-resistant bacteria [36–38]. The positive linear associations we observed between resistance to ceftriaxone and ceftazidime

**Table 1. Beta regression mixed models of the effect of antimicrobial consumption on the prevalence of beta-lactam and fluoroquinolone-resistant *E. coli* and *Klebsiella* spp. from bloodstream infections (the country was taken as a random effect).**

| Antimicrobial consumption (Variable) | Antimicrobial resistance (Outcome) | Regression estimates | | |
|---|---|---|---|---|
| | | Odds ratio | 95% Confidence interval | P-value |
| **Beta-lactam/Cephalosporins Consumption** | Bloodstream Ceftriaxone-R *E. coli* | 1.22 | 1.01–1.48 | **0.0395** |
| | Bloodstream Ceftazidime-R *E. coli* | 1.15 | 0.95–1.37 | 0.1315 |
| | Bloodstream Cefotaxime-R *E. coli* | 1.15 | 0.94–1.41 | 0.1810 |
| | Bloodstream Cefepime-R *E. coli* | 1.15 | 0.95–1.39 | 0.1470 |
| | Bloodstream Cefepime-R *Klebsiella* spp. | 1.09 | 0.89–1.33 | 0.404 |
| | Bloodstream Cefotaxime-R *Klebsiella* spp. | 1.16 | 0.94–1.45 | 0.168 |
| | Bloodstream Ceftazidime-R *Klebsiella* spp. | 1.20 | 0.98–1.47 | **0.0737** |
| | Bloodstream Ceftriaxone-R *Klebsiella* spp. | 1.11 | 0.99–1.25 | **0.0522** |
| **Quinolone Consumption** | Bloodstream Ciprofloxacin-R *E. coli* | 1.40 | 1.17–1.67 | **0.0002** |
| | Bloodstream Levofloxacin-R *E. coli* | 1.31 | 1.15–1.50 | **<0.0001** |
| | Bloodstream Ciprofloxacin-R *Klebsiella* spp. | 1.37 | 1.07–1.76 | **0.0118** |
| | Bloodstream Levofloxacin-R *Klebsiella* spp. | 1.31 | 1.02–1.68 | **0.0341** |

Statistically significant P-values (p<0.05) are in bold

**Table 2. Comparison of DDD per 1000 inhabitants per day of antimicrobial consumption between the antimicrobial drug ATC groups.** The comparison was adjusted for multiple comparisons using Tukey's method. Only marginally and statistically significant differences in group comparisons were presented.

| Antimicrobial group comparison | DDD difference | Standard error | P-value |
|---|---|---|---|
| **Beta-lactam antibacterials, penicillins (J01C)—Aminoglycoside antibacterials (J01G)** | 8.13 | 0.79 | **<0.0001** |
| **Other antibacterials (J01X)—Beta-lactam antibacterials, penicillins (J01C)** | -7.70 | 0.78 | **<0.0001** |
| **Intestinal antiinfectives (A07A)—Beta-lactam antibacterials, penicillins (J01C)** | -8.26 | 0.90 | **<0.0001** |
| **Sulfonamides and trimethoprim (J01E)—Beta-lactam antibacterials, penicillins (J01C)** | -7.44 | 0.80 | **<0.0001** |
| **Beta-lactam antibacterials, penicillins (J01C)—Agents against amoebiasis and other protozoal diseases (P01A)** | 7.06 | 0.78 | **<0.0001** |
| **Beta-lactam antibacterials, penicillins (J01C)—Amphenicols (J01B)** | 8.32 | 0.95 | **<0.0001** |
| **Tetracyclines (J01A)—Beta-lactam antibacterials, penicillins (J01C)** | -6.62 | 0.79 | **<0.0001** |
| **Quinolone antibacterials (J01M)—Beta-lactam antibacterials, penicillins (J01C)** | -6.28 | 0.78 | **<0.0001** |
| **Combinations of antibacterials (J01R)—Beta-lactam antibacterials, penicillins (J01C)** | -8.70 | 1.25 | **<0.0001** |
| **Macrolides, lincosamides and streptogramins (J01F)—Beta-lactam antibacterials, penicillins (J01C)** | -5.23 | 0.78 | **<0.0001** |
| **Other beta-lactam antibacterials (J01D)—Beta-lactam antibacterials, penicillins (J01C)** | -5.13 | 0.78 | **<0.0001** |
| **Other beta-lactam antibacterials (J01D)—Aminoglycoside antibacterials (J01G)** | 2.99 | 0.79 | **0.010** |
| **Macrolides, lincosamides and streptogramins (J01F)—Aminoglycoside antibacterials (J01G)** | 2.89 | 0.79 | **0.017** |
| **Other beta-lactam antibacterials (J01D)—Intestinal antiinfectives (A07A)** | 3.14 | 0.90 | **0.028** |
| **Macrolides, lincosamides and streptogramins (J01F)—Intestinal antiinfectives (A07A)** | 3.03 | 0.90 | **0.040** |
| **Other beta-lactam antibacterials (J01D)—Amphenicols (J01B)** | 3.19 | 0.95 | **0.044** |
| **Other beta-lactam antibacterials (J01D)—Other antibacterials (J01X)** | 2.57 | 0.78 | 0.051 |
| **Macrolides, lincosamides and streptogramins (J01F)—Amphenicols (J01B)** | 3.09 | 0.95 | 0.061 |
| **Other antibacterials (J01X)—Macrolides, lincosamides and streptogramins (J01F)** | -2.46 | 0.78 | 0.075 |

Statistically significant P-values (p<0.05) are in bold

in *E. coli* and *Klebsiella pneumoniae* and consumption of the beta-lactam antimicrobial agrees with previous studies which found increased antibiotic consumption to be associated with increased resistance in *E. coli* and *K. pneumoniae* [39, 40]. Our beta-regression model (Table 1) indicated a strong association between beta-lactam/cephalosporin antimicrobial consumption and resistance in bloodstream associated *E. coli* infection. Additionally, our study's findings on the positive linear associations between resistance to ciprofloxacin and levofloxacin in *E. coli* and *Klebsiella pneumoniae* and quinolone consumption are consistent with previous research. Okeke et al. emphasized in their study the link between fluoroquinolone usage and the emergence of resistance in *E. coli*, particularly in developing countries with less regulated access to antibiotics [41] which is also supported by beta-regression model (Table 1) indicating a stronger association between quinolone consumption and resistance found in bloodstream *E. coli* and *Klebsiella* infection. Ceftriaxone, ciprofloxacin, and levofloxacin fall in the "Watch" list of the WHO AWaRe antimicrobial classification which includes most of the highest priority antimicrobials that have an elevated risk of bacterial resistance selection, and ceftazidime is in the "Reserve" group which are last resort antimicrobial options for human treatment [17, 19]. The observed resistance to both the "Watch" and "Reserve" antimicrobial groups in *E. coli* and *Klebsiella pneumoniae* bloodstream infection is a huge concern that further negatively impacts the clinical outcomes of those infected because bloodstream infections are usually associated with poor clinical outcomes especially when there is delay diagnosis and treatment [42–44]. Because there is a scarcity of new antibiotics in development [45, 46], it is therefore pertinent that these antimicrobial classes be handled judiciously to avoid greater health disasters.

Furthermore, we observed a consistent and clear divide between developed and developing countries in terms of AMR in extended-spectrum cephalosporin and fluoroquinolone-

**Table 3. Comparison of DDD per 1000 inhabitants per day of antimicrobial consumption between the countries.** The comparison was adjusted for multiple comparisons using Tukey's method. Only marginally and statistically significant differences in group comparison were presented.

| Country comparison | DDD difference | standard error | P-value |
|---|---|---|---|
| **Uganda—Brunei Darussalam** | 5.01 | 1.05 | **0.0007** |
| **Uganda–Ivory Coast** | 4.87 | 1.05 | **0.0012** |
| **Brunei Darussalam–Iran** | -4.84 | 1.08 | **0.0019** |
| **Ivory Coast—Iran** | -4.70 | 1.08 | **0.0034** |
| **Uganda–Germany** | 4.61 | 1.06 | **0.0035** |
| **Uganda–Bhutan** | 4.57 | 1.05 | **0.0038** |
| **Uganda–Sweden** | 4.45 | 1.06 | **0.0061** |
| **Germany–Iran** | -4.44 | 1.07 | **0.0080** |
| **Bhutan—Iran** | -4.40 | 1.08 | **0.0097** |
| **Uganda–Peru** | 4.17 | 1.03 | **0.0112** |
| **Sweden—Iran** | -4.29 | 1.07 | **0.0132** |
| **Uganda–Benin** | 3.89 | 1.00 | **0.0197** |
| **Uganda–Denmark** | 4.07 | 1.06 | **0.0217** |
| **Peru–Iran** | -4.01 | 1.05 | **0.0245** |
| **United Kingdom–Uganda** | -3.86 | 1.03 | **0.0309** |
| **Uganda–Belgium** | 3.79 | 1.03 | **0.0372** |
| **Denmark–Iran** | -3.90 | 1.07 | **0.0431** |
| **Benin–Iran** | -3.73 | 1.04 | **0.0481** |
| **United Kingdom–Iran** | -3.70 | 1.05 | 0.0609 |
| **Belgium–Iran** | -3.63 | 1.05 | 0.0749 |

Statistically significant P-values (p < 0.05) are in bold

resistant *E. coli* and *Klebsiella pneumoniae* isolated from bloodstream infections and the consumption of beta-lactam/cephalosporins and quinolones among the participating countries (Figs 1–8). Developed countries such as Germany, Denmark, Sweden, the United Kingdom (UK), and Belgium were classified as low consumption-low resistance countries, while other countries fell into either the high consumption-high resistance or low consumption-high resistance categories. This clustering of countries in the correlation plots can be attributed to several reasons. All countries in the low consumption–low resistance in our correlation plot except Brunei Darussalam, are part of the European Union (EU), which has the most advanced AMR stewardship, surveillance, and monitoring system globally coordinated by the European Antimicrobial Resistance Surveillance Network (EARS-Net) [26, 27]. The UK has been actively engaged in combating antimicrobial resistance (AMR) through various stewardship programs. A notable initiative is the national antimicrobial stewardship strategy, developed in 2013 focused on enhancing knowledge and understanding of AMR, preserving the effectiveness of existing antimicrobials, and promoting the development of new antimicrobials. This initiative was succeeded by another 5-year One Health national action plan (NAP) in 2019 to control AMR using a broader multisectoral approach [47, 48]. These efforts have led to significant improvements in antimicrobial usage and clinical outcomes in the UK [47, 49]. Sweden has been at the forefront of combating antimicrobial resistance (AMR) with targeted and effective political commitments dating back from the mid-to-late 1990s and has a well-established and comprehensive multisectoral framework to address AMR, resulting in significant reductions in antimicrobial consumption and the prevalence of AMR over the years [50, 51]. Sweden's successful efforts in controlling AMC make it one of the lowest antimicrobial consumers in Europe[26, 50, 52]. Germany has made significant strides in combating antimicrobial

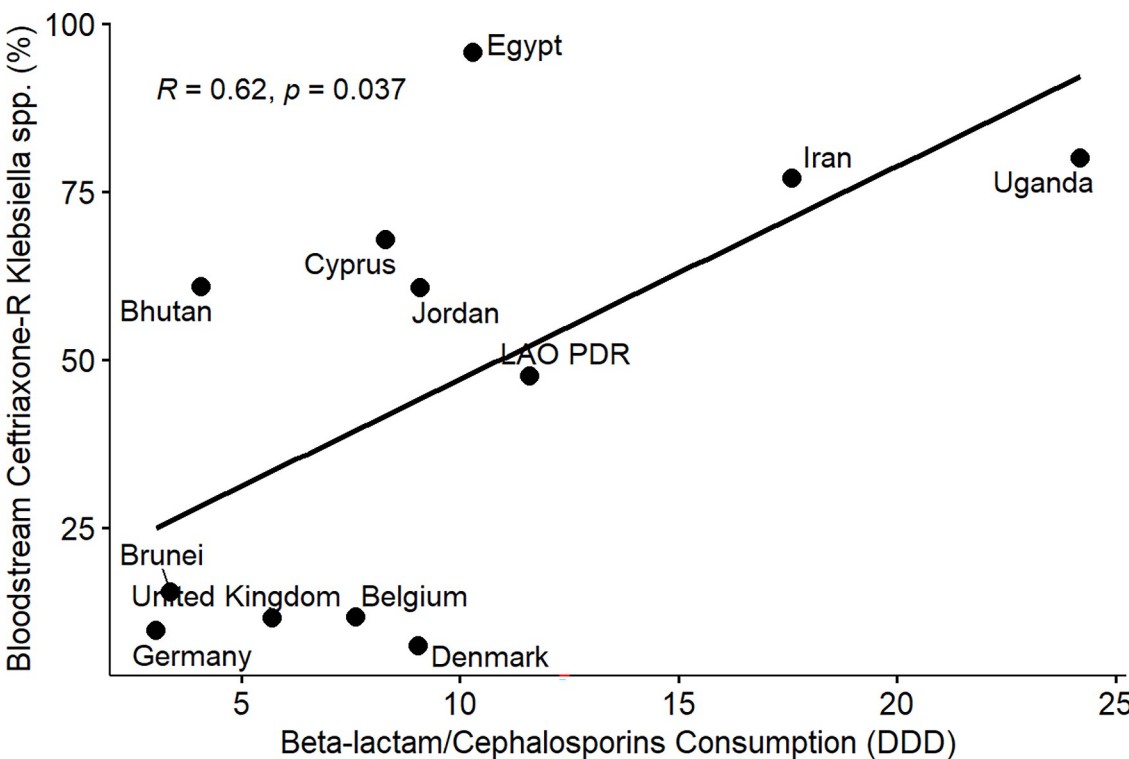

**Fig 3. Linear association and spearman correlation coefficients between bloodstream ceftriaxone-resistant *Klebsiella* spp. and cephalosporin consumption.**

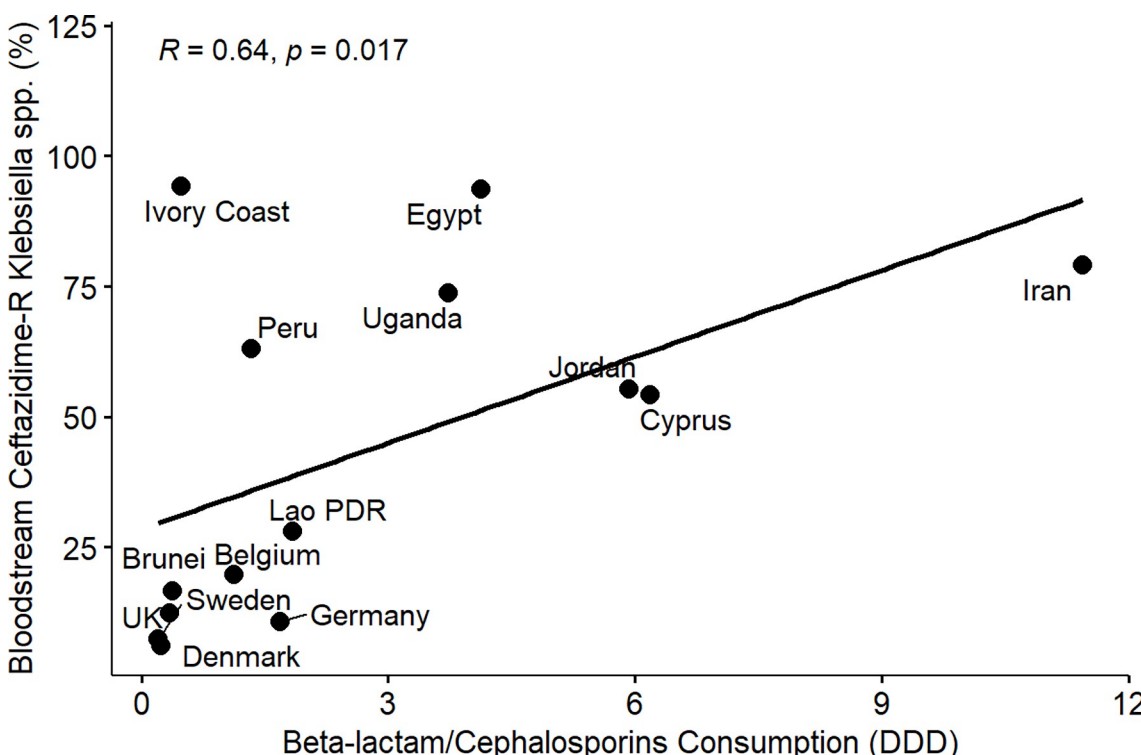

**Fig 4. Linear association and spearman correlation coefficients between bloodstream ceftazidime resistant *Klebsiella* spp. and cephalosporin consumption.**

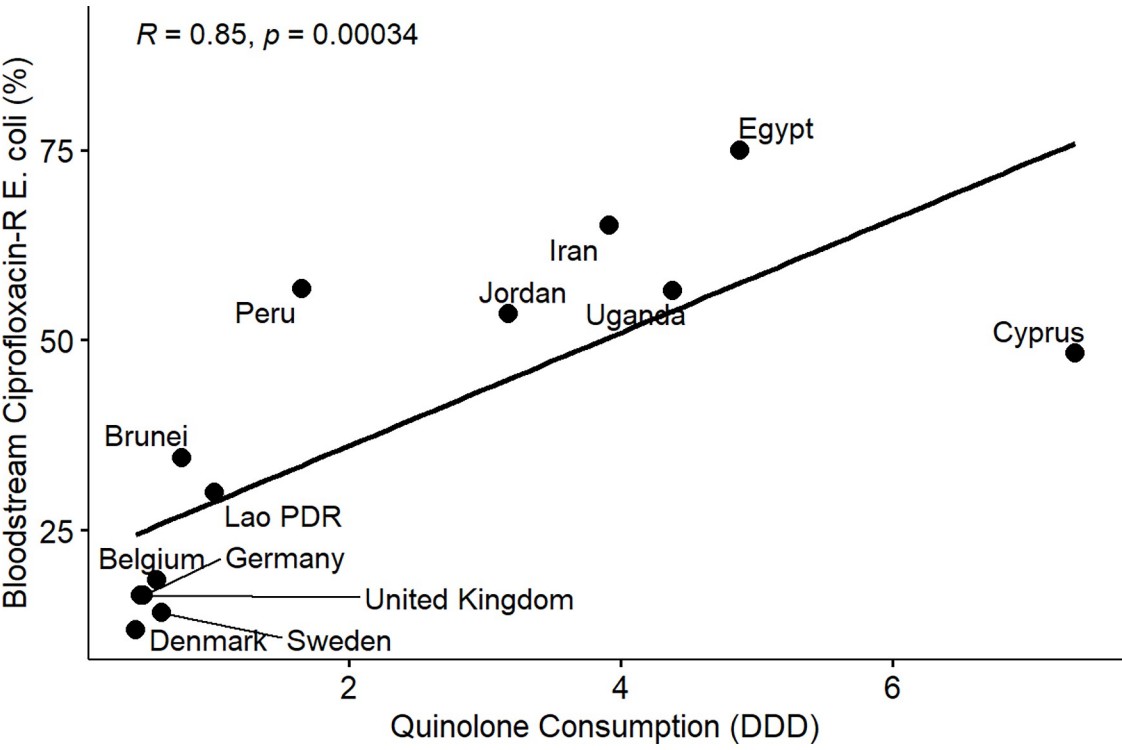

**Fig 5. Linear association and spearman correlation coefficients between bloodstream ciprofloxacin-resistant *E. coli* and quinolone consumption.**

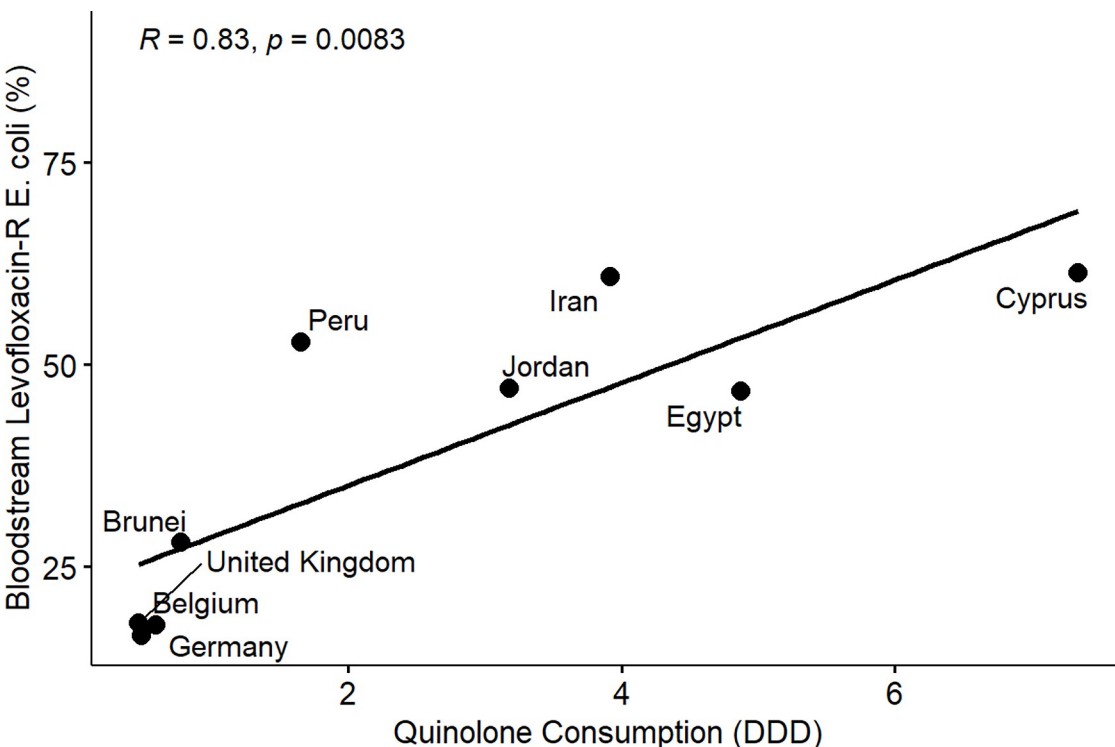

**Fig 6. Linear association and spearman correlation coefficients between bloodstream levofloxacin-resistant *E. coli* and quinolone consumption.**

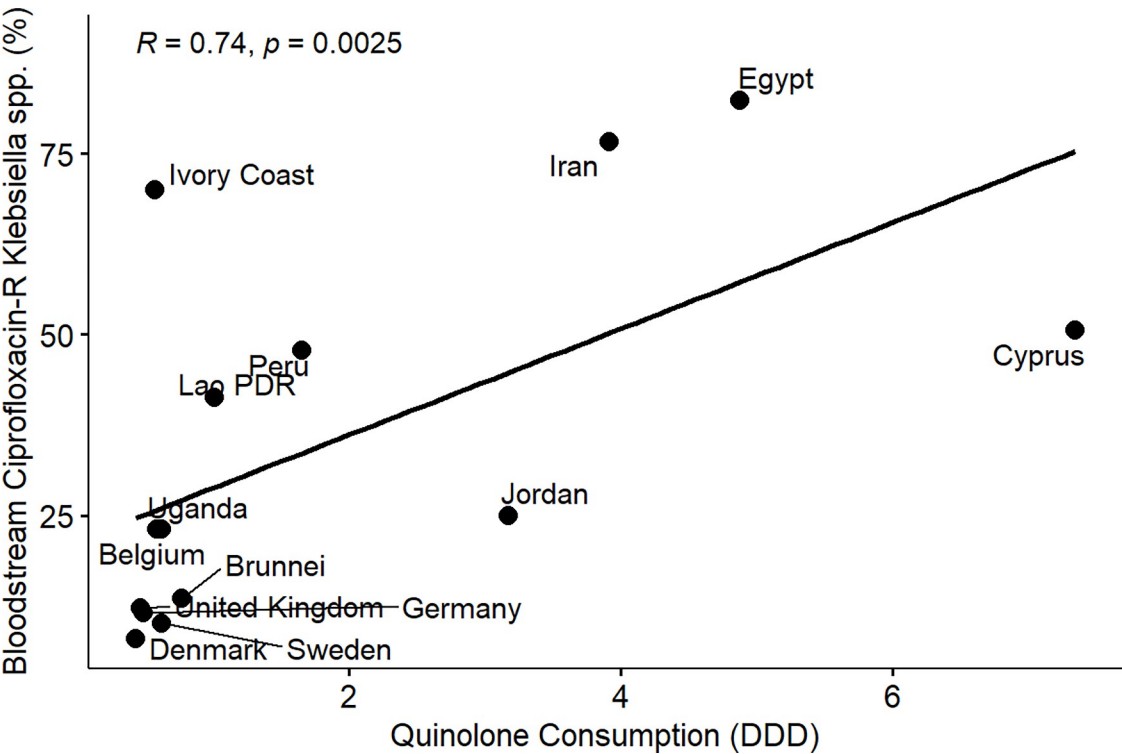

**Fig 7. Linear association and spearman correlation coefficients between bloodstream ciprofloxacin resistant *Klebsiella* spp. and quinolone consumption.**

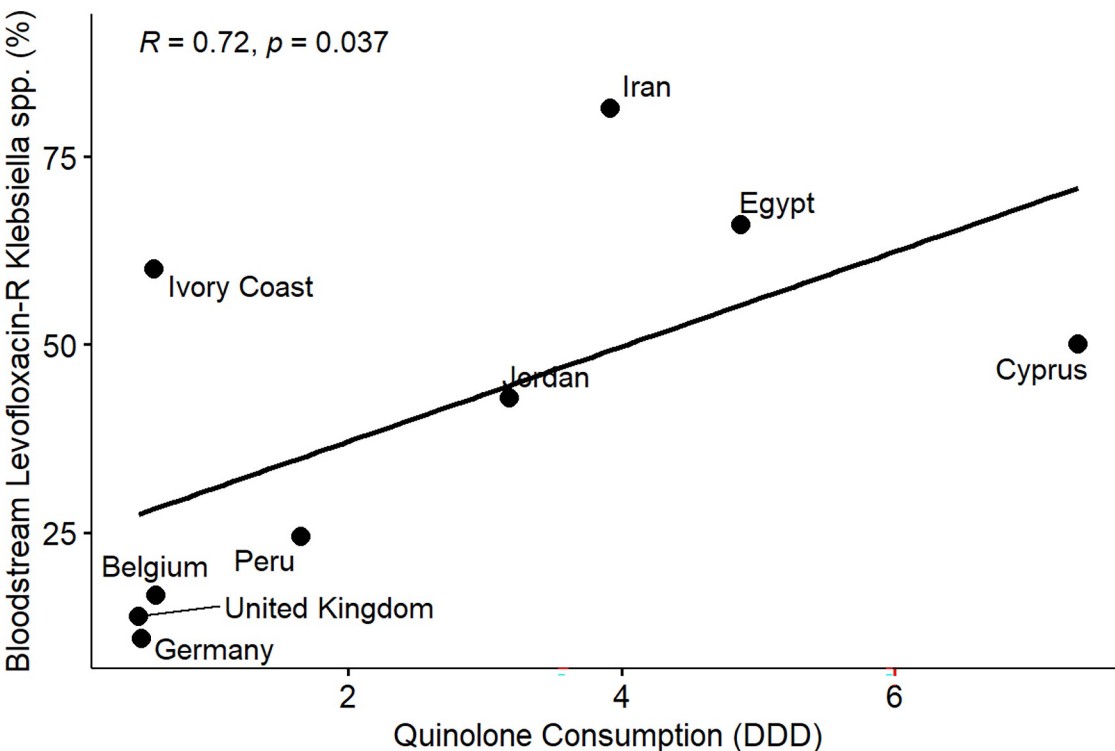

**Fig 8. Linear association and spearman correlation coefficients between bloodstream levofloxacin resistant *Klebsiella* spp. and quinolone consumption.**

resistance (AMR) through its national action plan, "DART 2020," which was set up in 2015 [53]. DART 2020 was a collaborative effort across multiple sectors to address AMR comprehensively and the implementation of this action plan has yielded positive results, as shown by the drastic reduction in antimicrobial consumption in Germany from 2016 to 2021, as reported by the EU annual AMC report [26]. This reduction in antimicrobial usage directly affected the AMR profile in the country during the same period and specifically, there was a notable decrease in combined resistance to third-generation cephalosporins, fluoroquinolones, and aminoglycosides in bacteria such as *E. coli* and *K. pneumoniae* [54, 55]. These achievements highlight the effectiveness of targeted efforts in combating AMR and serve as solid evidence of the importance of a coordinated approach to address the AMR global challenge. Aside from these European countries, Brunei, a south-eastern Asian country, also consistently falls under the low-consumption and low-resistance category in the correlation plots. Brunei Darussalam is a high-income country with an estimated population of less than 500, 000 which may be an advantage of an efficient and less pressurized healthcare system [56]. As part of Brunei's national action plan, there is a policy in place that restricts the availability of antimicrobials to prescription-only medicine [57]. Although, the observed low AMR is linked to low AMC generally, however, this may be an incomplete picture of what is truly obtainable in Brunei Darussalam, because AMR surveillance systems in the south-eastern Asia region were not fully developed and the AMC rates reported for Brunei Darussalam only covered the number of antimicrobials used solely in the public health care sector which is an incomplete coverage of AMC [27, 29].

The low consumption-high resistance category was consistently associated with countries such as Bhutan, Peru, and Ivory Coast which are low and middle-income countries LMICs. The multiple comparisons of AMC rates amongst countries from Table 3. showed that the Ivory Coast and Peru particularly had significantly lower ($p < 0.05$) consumption rates than Iran and Uganda which are classified under the high consumption- high resistance category in the correlation plots and Bhutan was also found to have significantly lower ($p < 0.05$) AMC compared to only Iran. From the GLASS report, the AMC records sources for Peru and Bhutan have their AMC data from hospitals, pharmacies, and central drug stores which usually give a better estimate of AMC compared to the Ivory Coast records that were solely from wholesale records [29, 58]. In Bhutan, access to antimicrobials for systemic use in humans is carefully regulated by the government and dependent on prescriptions from pharmacies in healthcare facilities controlled by the government [29, 58]. Although, a study on trends in antimicrobial consumption in Bhutan, reported that overall AMC had consistently increased from 2017–2019 which could contribute to increased bacteria resistance, however, AMC in Bhutan was still lower than the EU average [58]. The 2021 WHO Tripartite AMR Country Self-Assessment Survey (TrACSS) for Bhutan indicates progress in AMR and AMC surveillance in humans and animals. However, there is a need to enhance capacity for optimizing antimicrobial use in plant production and reducing infection rates.[59]. These may be possible reasons for elevated and persisting resistance despite significant measures to reduce human consumption of antimicrobials. Furthermore, information relating to AMR specifically beta-lactam and quinolone resistance in Bhutan is sparse in the literature which limits our view of AMR dynamics in Bhutan.

Similarly, Peru had well developed AMR and AMC surveillance and monitoring system in humans and also a developed capacity to reduce the incidence of infection compared to Bhutan, however, it had no capacity for AMR surveillance in animals and also a poor capacity in optimizing AMC in animals and for plant production from the 2021 TrACSS report [60], which may be a major driving factor for AMR in Peru. Indiscriminate and overuse of antimicrobials in livestock as prophylaxis and growth promotion could be a major contributing

factor to the relatively high resistance, a study showed a highly significant correlation in the resistance to fluoroquinolones and third-generation cephalosporins between *E. coli* from human bloodstream infection and *E. coli* isolates from poultry, pigs, and cattle and this supports our assertion that asides from human AMC, livestock AMC could also be a significant AMR driver in humans [61, 62]. For Ivory Coast, the 2021 TrACSS report showed that AMR and AMC surveillance in human animal, and plant production is not well developed [63], and the observed low resistance could be due to the possibility of data underreporting. Surveillance records from LMICs are still largely inconsistent due to data underreporting challenges [20, 29, 64]. Accessing antimicrobials in LMICs at the community level poses significant challenges, one key challenge is the widespread availability of antimicrobials without proper authorization or prescription, known as non-licensed dispensing [65–67]. Batista A. D et al., systematic review study found that more than 60% of antimicrobials were dispensed without prescription in 83% (49 of 59) studies done in LMICs [68]. In addition, the notable prevalence of non-licensed dispensing of antimicrobials is fueled by the practice of self-medication, a study reported that about 60% of individuals interviewed had bought antimicrobials for self-medication from selected private pharmacies in Abidjan, Ivory Coast [69]. This behavior of self-medication is motivated by factors such as convenience, low cost, and the belief that it is less time-consuming compared to seeking healthcare services [65]. However, unregulated access to antimicrobials can contribute to inappropriate and excessive usage, which in turn fuels the development and transmission of antimicrobial resistance. AMC data from these practices cannot be easily accounted for and usually go unreported making the average consumption data reported lower than actual values. In addition, some studies have found beta-lactams/ cephalosporins and fluoroquinolones as antimicrobials often used for self-medication which may be an explanation for the observed low-consumption and high resistance found in these LMICs [68, 70]. Additionally, the paradox of LMICs exhibiting high resistance despite low antimicrobial consumption could be attributed to a lack of resources for implementing interventions such as infection control [71–73]. This underscores the significance of considering socio-economic factors and resource constraints in addressing antimicrobial resistance (AMR) effectively.

Cyprus, Iran, Egypt, Jordan, and Uganda stand out consistently in the high consumption-high resistance category in all correlation plots which underscores the fact that AMR knows no boundaries, affecting both low and high-income countries alike, and its magnitude is inextricably linked to the extent of AMC, although, low income and middle-income countries are more affected [74]. Cyprus is under the European Union (EU) which has the best regional efforts towards surveillance, however, several studies have shown that Cyprus is one of the top users of antibiotics in both community and hospital settings within Europe [54, 75, 76]. Within hospitals in Cyprus, extended-spectrum antibiotics such as third and fourth-generation cephalosporins, fluoroquinolones, and carbapenems make up over 50% of the antibiotics used, and community antimicrobial consumption was driven by easy access to antimicrobials and lack of knowledge of AMC indications and AMR [76–78]. In 2014, WHO reported high *E. coli* resistance to third-generation cephalosporin (41%) and Fluoroquinolones (54%) and *Klebsiella pneumonia* to third-generation cephalosporin (48%), and Carbapenems (54%) in Iran [27]. Also, extremely high rates of resistance have been reported constantly in Iran in all WHO GLASS reports, although it had few surveillance sites ranging from 12 in the 2017–2018 WHO early implementation GLASS reports to 16 in the 2021 GLASS report [20, 28, 79]. However, other independent studies also confirmed high bacterial resistance rates in Iran which have been attributed to the misuse of antimicrobials [80–83].

Iran is an upper- and middle-income level country and has significant endeavors towards promoting rational use of antimicrobials, however, irrational, and unlawful prescription of

antimicrobials, and self-medication still largely persists [84–87]. The multiple comparisons of our beta regression model (Table 3) showed that Iran had significantly higher (p < 0.05) AMC rates compared to any other country other than Uganda, and this confirms AMC is a major precipitator of AMR in Iran. Most Low-income, and Low and middle-income countries such as Egypt, Jordan, and Uganda, have high disease burdens and high resistance rates due to poor healthcare infrastructures, scarce economic resources, little or no legislative enforcement toward controlling antimicrobial access and use, poor education, and AMR awareness and ineffective AMR supply chain [23, 27, 71, 88–91]. Uganda stands out amongst the 3 from our analysis as having significantly higher rates of AMC compared to any other country (Table 3), which points out AMC as an important driver of AMR in Uganda. Although, limited information exists about the actual magnitude of the antimicrobial resistance (AMR) issue in LICs and LMICs due to inadequate and unreliable AMR surveillance systems, however, consistently high AMR resistance rates have been reported in LICs and LMICs in all WHO GLASS reports over the years and several independent studies [20, 28]. Also, the observed high consumption and high resistance could be attributed to an inefficient supply chain of antimicrobials leading to the inappropriate use of alternative drugs and escalating the risk of antimicrobial resistance [91, 92].

Focusing on global AMC, based on antimicrobial type/class, we noted a significant contrast of high daily usage of beta-lactam-penicillins and cephalosporins compared to other types of antimicrobials. These classes of antimicrobials, particularly beta-lactams-penicillins, constitute more than 60% of antimicrobials being produced, also, they are considered an empirical therapy for most bacterial infections, especially in developing countries with poor health and diagnostics facilities and a high prevalence of self-medication and non-prescription antimicrobial consumption [36, 93–95]. The overall availability and accessibility of these antimicrobial classes contribute to the higher consumption rates [91]. The insufficient awareness and understanding of antimicrobial usage and indications contribute to the indiscriminate use of beta-lactams (penicillin) and cephalosporins for treating conditions like colds, flu, and other viral infections, exacerbating the inappropriate utilization of these antimicrobial classes.[93, 96, 97].

Our study has some limitations that warrant consideration. Firstly, we acknowledge the constraint of having AMC data available for only 26 countries, with limited complete coverage in AMR and AMC data reported. This may result in a skewed representation that does not fully capture global patterns. Secondly, our focus on *E. coli* and *K. pneumoniae* associated with bloodstream infection represents only a fraction of resistant bacteria reported in the 2022 GLASS report. As such, it may not provide a comprehensive view of the relationship between AMC and AMR across various bacterial strains. Thirdly, it's important to note that the GLASS data utilized in our study is based solely on 2020 data, which may not accurately reflect recent shifts in antimicrobial consumption or resistance trends. Despite these limitations, our research provides valuable insights into the intricate relationship between antimicrobial consumption and resistance at both national and global scales. These insights have the potential to inform and guide the development of effective global public health policies and intervention strategies aimed at combating the pervasive threat of antimicrobial resistance.

## Conclusion

These observations underscore the necessity for enhanced antimicrobial stewardship (AMS) and improved surveillance systems in both developed and developing countries to curtail the emergence and dissemination of antimicrobial resistance. AMS is a critical tool to combat AMR and it involves a well-organized set of actions that advocates for and promotes responsible use of antimicrobials which could be at an individual, national, and global level and

encompassing various aspects such as human health, animal health, and environmental considerations [49, 74, 98].

Our observation in this study clearly shows that reduced AMC is strongly associated with reduced AMR rates and most developed countries like the United Kingdom, United States, and Sweden have well-developed and robust AMS programs through national action plans and strategies, enforcing regulatory policies for AMC, multisectoral and one-health collaborations, and constantly improving AMR/AMC surveillance systems [49–51, 99], and this has led to the overall reductions of both AMC and AMR rates as opposed to most developing countries without well-developed AMS and surveillance programs. Due to globalization as a result of travel and trade, resistant pathogens can easily spread from one country to another [100, 101], several studies have shown that individuals who traveled from high-income countries to certain low and middle-income countries (LMICs) returned with increased AMR bacteria resistance after screening before and after travels [102–104]. Therefore, increasing efforts must be targeted towards strengthening AMS and surveillance systems, especially in LMICs and LICs.

This study also highlights the significance of integrated surveillance of AMR which encompasses integration and analysis of both AMR and AMC surveillance data at the one health interface [51, 89, 105]. Analysis of both AMR and AMC data provides more insight into the trends of AMR, and the impact of AMC on AMR which improves policy action and planning for AMS programs [106]. Although our study focuses on only human AMC and AMR data, integrating and analyzing AMC and AMR in the One Health interface is very critical to properly understanding the drivers of AMR [31], and studies have established that AMC and AMR in animals and environment could also drive AMR in humans [38, 107].

## Acknowledgments

The data used in this study were extracted from the World Health Organization's Global Antimicrobial Resistance and Use Surveillance System (GLASS) report published for 2022 [20].

## Author Contributions

**Conceptualization:** Babafela Awosile.

**Data curation:** Samuel Ajulo.

**Formal analysis:** Babafela Awosile.

**Investigation:** Samuel Ajulo, Babafela Awosile.

**Methodology:** Babafela Awosile.

**Supervision:** Babafela Awosile.

**Writing – original draft:** Samuel Ajulo, Babafela Awosile.

**Writing – review & editing:** Samuel Ajulo, Babafela Awosile.

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
