## [Decision Letter · Decision Letter 0]

21 Nov 2023

PONE-D-23-32918Global Antimicrobial Resistance and Use Surveillance System (GLASS 2022): Investigating the relationship between antimicrobial resistance and antimicrobial consumption data across the participating countries.PLOS ONE

Dear Dr. Babafela,

Thank you for submitting your manuscript to PLOS ONE. After careful consideration, we feel that it has merit but does not fully meet PLOS ONE’s publication criteria as it currently stands. Therefore, we invite you to submit a revised version of the manuscript that addresses the points raised during the review process.

We look forward to receiving your revised manuscript.

Kind regards,

Balew Arega Negatie, Msc,MD

Academic Editor

PLOS ONE

Reviewers' comments:

Reviewer's Responses to Questions

**Comments to the Author**

1. Is the manuscript technically sound, and do the data support the conclusions?

Reviewer #1: Yes

Reviewer #2: Yes

2. Has the statistical analysis been performed appropriately and rigorously? 

Reviewer #1: Yes

Reviewer #2: Yes

3. Have the authors made all data underlying the findings in their manuscript fully available?

Reviewer #1: Yes

Reviewer #2: Yes

4. Is the manuscript presented in an intelligible fashion and written in standard English?

Reviewer #1: Yes

Reviewer #2: No

5. Review Comments to the Author

Reviewer #1: The manuscript is well written, and the findings will add to the body of evidence on association of antimicrobial resistance with antimicrobial consumption (AMC). However, drawing a direct correlation of the AMCs of participating countries with the resistance may not necessarily provide an accurate picture of the actual relationship owing to the varying data sources and the comprehensiveness thereof. This is evident from finding of country groupings into low consumption-low resistance, high-consumption high resistance, low consumption - high resistance and so forth. The contribution of the confounding factors may be elaborated under the study limitations.

As for the findings of statistically significant higher consumption of beta lactam penicillin and cephalosporins amongst the ATC groups, the GLASS report is self-explanatory even without application of any statistical method. Suggested to focus on the findings of correlation between resistance and consumption.

Abilities of the countries to implement antimicrobial stewardship interventions including infection control in the healthcare facilities would also have a huge bearing on the AMR and the paradox of LICs having high resistance even with low consumption could be because of lack of resources to implement this intervention. This may also be touched upon under the discussion.

Line 188- 199: Is the word "recovery" appropriate in the context of 'odds of isolation/detection of resistant organism' reported by the countries? Suggested to replace with a suitable alternative.

Reviewer #2: This article provides crucial information to link the reported AMR data with the AMC data in different countries and regions based on the reported GLASS data, focusing on beta-lactam/cephalosporin and quinolones AMC data and beta-lactam/cephalosporin and quinolones resistant E. coli and Klebsiella pneumoniae and also explore the relationship and differences between the AMC data reported among the participating countries.

The authors performed critical analysis to gain better insight into GLASS data and makes inferences for integrated analysis of AMR and AMC data and to inform AMR intervention strategies at a global scale.

This study has great prospect to be useful to the international AMR community but needs to provide certain clarifications with minor revisions.

- The authors need to provide acknowledgements especially for the data source or any support received.

- The authors need to provide stronger rationale/ justification for limiting their analysis to blood stream infections and specifically E. coli and K. pneumonia, and also for the selected countries reported.

- Study limitations should be clearly articulated and separated from the conclusion session.

- The entire manuscript will benefit from English language editing for punctuations and use of words.

- Intext referencing style needs to be adjusted and consistent.

- Table 1: I assume that the reason for the bolding is to show significance, if so, then 0.0737 is not significant by your assumption of 0.05 significance level

- The discussion session was adequate with robust deliberation on the study findings citing relevant literatures.

Minor revisions

-Abstract Line 33: State the actual p value, Line 37-38; DDD, ATC write in full at first appearance

- Pg 7 line 148: Add a full-stop after pneumonia

- Pg 10, line 227: “ consistent with previous research…” please provide reference for this statement

-Pg 12 line 258: “promoting the development of new antimicrobials…” Add a full stop before. …..This initiative

- Line 293 and 294: Add actual p value and not ≤0.05

- Pg 13&14 Line 302-308: “Also the 2021……..” this sentence is too ling and unwinding and lacks clarity. Kindly separate into two sentences

-Pg 16Line 355-357: This statement is not clear. Are you inferring that the reduced number of reporting sites was due to underreporting or underreporting was responsible for fewer sites in Iran. Either way, I fail to see the connection.

-Pg 17 Line 388: Remove “application of “ to read as “such that beta-lactam…..” or rephrase the entire sentence for clarity

-Line 411: Add a full stop before “therefore”

6. PLOS authors have the option to publish the peer review history of their article (what does this mean?). If published, this will include your full peer review and any attached files.

Reviewer #1: No

Reviewer #2: **Yes: **Emelda Eberechukwu Chukwu

---

## [Author Response · Author response to Decision Letter 0]

4 Jan 2024

PONE-D-23-32918

Global Antimicrobial Resistance and Use Surveillance System (GLASS 2022): Investigating the relationship between antimicrobial resistance and antimicrobial consumption data across the participating countries.

PLOS ONE

We thank the reviewers for their suggestions. These suggestions have helped to improve the quality and readability of our manuscript. Thanks

Reviewers' comments:

Reviewer's Responses to Questions 

Comments to the Author

1. Is the manuscript technically sound, and do the data support the conclusions?

Reviewer #1: Yes

Reviewer #2: Yes

2. Has the statistical analysis been performed appropriately and rigorously? 

Reviewer #1: Yes

Reviewer #2: Yes

3. Have the authors made all data underlying the findings in their manuscript fully available?

The PLOS Data policy requires authors to make all data underlying the findings described in their manuscript fully available without restriction, with rare exception (please refer to the Data Availability Statement in the manuscript PDF file). The data should be provided as part of the manuscript or its supporting information or deposited to a public repository. For example, in addition to summary statistics, the data points behind means, medians and variance measures should be available. If there are restrictions on publicly sharing data—e.g. participant privacy or use of data from a third party—those must be specified.

Reviewer #1: Yes

Reviewer #2: Yes

4. Is the manuscript presented in an intelligible fashion and written in standard English?

Reviewer #1: Yes

Reviewer #2: No

5. Review Comments to the Author

Reviewer #1: The manuscript is well written, and the findings will add to the body of evidence on association of antimicrobial resistance with antimicrobial consumption (AMC). However, drawing a direct correlation of the AMCs of participating countries with the resistance may not necessarily provide an accurate picture of the actual relationship owing to the varying data sources and the comprehensiveness thereof. This is evident from finding of country groupings into low consumption-low resistance, high-consumption high resistance, low consumption - high resistance and so forth. The contribution of the confounding factors may be elaborated under the study limitations.

As for the findings of statistically significant higher consumption of beta lactam penicillin and cephalosporins amongst the ATC groups, the GLASS report is self-explanatory even without application of any statistical method. Suggested to focus on the findings of correlation between resistance and consumption.

Abilities of the countries to implement antimicrobial stewardship interventions including infection control in the healthcare facilities would also have a huge bearing on the AMR and the paradox of LICs having high resistance even with low consumption could be because of lack of resources to implement this intervention. This may also be touched upon under the discussion.

Response: Thank you so much for pointing this out. We have been able to point out with appropriate references, that the lack of resources to implement infection control intervention is a factor for the observed paradox of high resistance and low consumptions in LMICs.

Line 188- 199: Is the word "recovery" appropriate in the context of 'odds of isolation/detection of resistant organism' reported by the countries? Suggested to replace with a suitable alternative.

Response: Thank you for your observations and comments to ensure the clarity of this manuscript. We have replaced “recovery” with isolation which appear appropriate in this situation. 

Reviewer #2: This article provides crucial information to link the reported AMR data with the AMC data in different countries and regions based on the reported GLASS data, focusing on beta-lactam/cephalosporin and quinolones AMC data and beta-lactam/cephalosporin and quinolones resistant E. coli and Klebsiella pneumoniae and also explore the relationship and differences between the AMC data reported among the participating countries.

The authors performed critical analysis to gain better insight into GLASS data and makes inferences for integrated analysis of AMR and AMC data and to inform AMR intervention strategies at a global scale.

This study has great prospect to be useful to the international AMR community but needs to provide certain clarifications with minor revisions.

- The authors need to provide acknowledgements especially for the data source or any support received.

Response: Thank you for your observation and comment on this, we have now provided proper acknowledgment for the manuscript. 

- The authors need to provide stronger rationale/ justification for limiting their analysis to blood stream infections and specifically E. coli and K. pneumonia, and also for the selected countries reported.

Response: Thank you for your comments. The reason for our focus on blood stream infections and specifically E. coli and Klebsiella pneumoniae. was partly outlined in lines 111-117, specifically because higher level of resistance associated with klebsiella pneumoniae globally and the worldwide spread of carbapenems-resistant Enterobacterales. We have further buttressed our rationale of limiting our analysis blood stream infections and specifically E. coli and Klebsiella pneumoniae because we are more likely to see a distinct more distinct relationship between antimicrobial use and resistance. 

- Study limitations should be clearly articulated and separated from the conclusion session.

Response: Thank you for your observation and comments on this point. The limitations of this study have been separated from the conclusion. It has also been updated and rephrased for better clarity. 

- The entire manuscript will benefit from English language editing for punctuations and use of words.

Response: We have worked on editing this manuscript to ensure clarity and proper punctuation.

- Intext referencing style needs to be adjusted and consistent.

Response: Thank you for your comment, appropriate changes have been made with the referencing 

- Table 1: I assume that the reason for the bolding is to show significance, if so, then 0.0737 is not significant by your assumption of 0.05 significance level

Response: Thank you for your observation and comments on this point. We have adjusted the table to reflect these changes and we also made some changes in the result section lines 194-205 to reflect that only p value < 0.5 were considered in our analysis.

- The discussion session was adequate with robust deliberation on the study findings citing relevant literatures.

Minor revisions

-Abstract Line 33: State the actual p value, Line 37-38; DDD, ATC write in full at first appearance

Response: Thank you for our observations and comments. For our analysis the p value was set at < 0.5, This p value varied in the different analysis, however, only the results showing p value at < 0.5 were considered and this has been updated already in the manuscript. The DDD, ATC full meaning has also been updated in the abstract section.

- Pg 7 line 148: Add a full-stop after pneumonia

Response: Thank you for our observations and comments. We have updated this full-stop punctuation mark.

- Pg 10, line 227: “consistent with previous research…” please provide reference for this statement

Response: Thank you for our observations and comments. The studies that were referred to in the statement were cited immediately (line 225), citation number 36-38.

-Pg 12 line 258: “promoting the development of new antimicrobials…” Add a full stop before. …..This initiative

Response: Thank you for our observations and comments. We have updated this full-stop punctuation mark.

- Line 293 and 294: Add actual p value and not ≤0.05 

Response: Thank you for our observations and comments. We added the reference to the table showing complete details about the individual p values. Since it is multiple comparison adding several different p-values that have been clearly outline in the result table might look clumsy to the readers, however, that was why we stated that p value < 0.05 as a way of summarizing the observations.

- Pg 13&14 Line 302-308: “Also the 2021……..” this sentence is too ling and unwinding and lacks clarity. Kindly separate into two sentences

Response: Thank you for our observations and comments. This has been clearly rewritten and separated into two sentences.

-Pg 16Line 355-357: This statement is not clear. Are you inferring that the reduced number of reporting sites was due to underreporting or underreporting was responsible for fewer sites in Iran. Either way, I fail to see the connection.

Response: Thank you for our observations and comments. This statement has been reviewed and edited to clearly explain the connection. The connection is that although Iran had few surveillance sites, other independent studies within the countries have confirmed high resistance rates, attributing it to misuse of antimicrobials.

-Pg 17 Line 388: Remove “application of “ to read as “such that beta-lactam…..” or rephrase the entire sentence for clarity

Response: Thank you for our observations and comments. This statement has been reviewed and paraphrased to ensure clarity.

-Line 411: Add a full stop before “therefore”

Response: Thank you for our observations and comments. We have updated this full-stop punctuation mark.

6. PLOS authors have the option to publish the peer review history of their article (what does this mean?). If published, this will include your full peer review and any attached files.

Do you want your identity to be public for this peer review? For information about this choice, including consent withdrawal, please see our Privacy Policy.

Reviewer #1: No

Reviewer #2: Yes: Emelda Eberechukwu Chukwu

---

## [Decision Letter · Decision Letter 1]

16 Jan 2024

Global Antimicrobial Resistance and Use Surveillance System (GLASS 2022): Investigating the relationship between antimicrobial resistance and antimicrobial consumption data across the participating countries.

PONE-D-23-32918R1

Dear Dr.Samuel

We’re pleased to inform you that your manuscript has been judged scientifically suitable for publication and will be formally accepted for publication once it meets all outstanding technical requirements.

Kind regards,

Balew Arega Negatie, Msc,MD

Academic Editor

PLOS ONE

Additional Editor Comments (optional):

Reviewers' comments:

Reviewer's Responses to Questions

**Comments to the Author**

1. If the authors have adequately addressed your comments raised in a previous round of review and you feel that this manuscript is now acceptable for publication, you may indicate that here to bypass the “Comments to the Author” section, enter your conflict of interest statement in the “Confidential to Editor” section, and submit your "Accept" recommendation.

Reviewer #1: All comments have been addressed

Reviewer #2: All comments have been addressed

2. Is the manuscript technically sound, and do the data support the conclusions?

Reviewer #1: Yes

Reviewer #2: Yes

3. Has the statistical analysis been performed appropriately and rigorously? 

Reviewer #1: Yes

Reviewer #2: Yes

4. Have the authors made all data underlying the findings in their manuscript fully available?

Reviewer #1: Yes

Reviewer #2: Yes

5. Is the manuscript presented in an intelligible fashion and written in standard English?

Reviewer #1: Yes

Reviewer #2: Yes

6. Review Comments to the Author

Reviewer #1: My comments have been addressed and necessary changes have been incorporated as suggested. I have no further comments to make. .

Reviewer #2: (No Response)

7. PLOS authors have the option to publish the peer review history of their article (what does this mean?). If published, this will include your full peer review and any attached files.

Reviewer #1: No

Reviewer #2: **Yes: **Chukwu Emelda Eberechukwu

---

## [Editor Report · Acceptance letter]

29 Jan 2024

PONE-D-23-32918R1 

PLOS ONE

Dear Dr. Awosile, 

I'm pleased to inform you that your manuscript has been deemed suitable for publication in PLOS ONE. Congratulations! Your manuscript is now being handed over to our production team.

Kind regards, 

on behalf of

Dr. Balew Arega Negatie 

Academic Editor

PLOS ONE